# Blinatumomab and Inotuzumab Ozogamicin Sequential Use for the Treatment of Relapsed/Refractory Acute Lymphoblastic Leukemia: A Real-Life Campus All Study

**DOI:** 10.3390/cancers15184623

**Published:** 2023-09-19

**Authors:** Nicola Stefano Fracchiolla, Mariarita Sciumè, Cristina Papayannidis, Antonella Vitale, Sabina Chiaretti, Mario Annunziata, Fabio Giglio, Prassede Salutari, Fabio Forghieri, Davide Lazzarotto, Monia Lunghi, Annalisa Imovilli, Barbara Scappini, Massimiliano Bonifacio, Michelina Dargenio, Carmela Gurrieri, Elisabetta Todisco, Marzia Defina, Maria Ilaria Del Principe, Patrizia Zappasodi, Marco Cerrano, Lidia Santoro, Elena Tagliaferri, Enrico Barozzi, Pasquale De Roberto, Marta Canzi, Elisa Buzzatti, Chiara Sartor, Francesco Passamonti, Robin Foà, Antonio Curti

**Affiliations:** 1Hematology Unit, Fondazione IRCCS Ca’ Granda Ospedale Maggiore Policlinico, 20122 Milan, Italy; mariarita.sciume@policlinico.mi.it (M.S.); elena.tagliaferri@policlinico.mi.it (E.T.); pasquale.deroberto@policlinico.mi.it (P.D.R.); marta.canzi@unimi.it (M.C.); francesco.passamonti@policlinico.mi.it (F.P.); 2IRCCS Azienda Ospedaliero-Universitaria di Bologna, Istituto di Ematologia “L. & A. Seràgnoli”, 40138 Bologna, Italy; cristina.papayannidis@unibo.it (C.P.); chiara.sartor1988@gmail.com (C.S.); antonio.curti2@unibo.it (A.C.); 3Hematology, Department of Translational and Precision Medicine, Sapienza University, 00185 Rome, Italy; vitale@bce.uniroma1.it (A.V.); sabina.chiaretti@uniroma1.it (S.C.); rfoa@bce.uniroma1.it (R.F.); 4Hematology Unit, Azienda Ospedaliera Cardarelli, 11411 Naples, Italy; annunziatam@libero.it; 5Division of Onco-Hematology, European Institute of Oncology, IRCCS, 20141 Milan, Italy; fabio.giglio@ieo.it; 6Hematology Unit, Ospedale Civile Santo Spirito, 65100 Pescara, Italy; prassede.salutari@asl.pe.it; 7Department of Medical and Surgical Sciences, Section of Hematology, Azienda Ospedaliero-Universitaria di Modena, University of Modena and Reggio Emilia, 41125 Modena, Italy; fabio.forghieri@unimore.it; 8Division of Hematology, University Hospital-ASUFC, 33100 Udine, Italy; davide.lazzarotto@asufc.sanita.fvg.it; 9Division of Hematology, Department of Translational Medicine, AOU Maggiore della Carità, Università del Piemonte Orientale, 13100 Novara, Italy; monia.lunghi@med.uniupo.it; 10Hematology Unit, Azienda Unità Sanitaria Locale-IRCCS, 42123 Reggio Emilia, Italy; annalisa.imovilli@ausl.re.it; 11Hematology Unit, Department of Experimental and Clinical Medicine, Azienda Ospedaliero-Universitaria Careggi, University of Florence, 50121 Florence, Italy; scappinib@aou-careggi.toscana.it; 12Department of Medicine, Section of Hematology, University of Verona, 37129 Verona, Italy; massimiliano.bonifacio@univr.it; 13Hematology and Stem Cell Transplantation Unit, Vito Fazzi Hospital, 73100 Lecce, Italy; miviforina@tiscali.it; 14Dipartimento Strutturale Aziendale Medicina, University of Padova, 35122 Padua, Italy; carmela.gurrieri@sanita.padova.it; 15Ospedale di Busto Arsizio, ASST Valle Olona, 21052 Busto Arsizio, Italy; elisabetta.todisco@asst-valleolona.it; 16Hematology Unit, University of Siena, Azienda Ospedaliera Universitaria Senese, 53100 Siena, Italy; marzia.defina@ao-siena.toscana.it; 17Hematology Unit, Department of Biomedicina and Prevention, Tor Vergata University, 00133 Rome, Italy; dlpmlr00@uniroma2.it (M.I.D.P.); buzzattielisa@gmail.com (E.B.); 18Department of Hematology Oncology, Division of Hematology, Fondazione IRCCS Policlinico San Matteo, 27100 Pavia, Italy; p.zappasodi@smatteo.pv.it; 19Division of Hematology, A.O.U. Città della Salute e della Scienza, 10126 Turin, Italy; cerranomarco@gmail.com; 20Struttura Complessa di Ematologia e Trapianto Emopoietico, A.O.S.G. Moscati, 83100 Avellino, Italy; lidiasantoro63@libero.it; 21Department of Oncology and Hemato-Oncology, University of Milan, 20122 Milan, Italy; enrico.barozzi@unimi.it

**Keywords:** blinatumomab, inotuzumab ozogamicin, immunotherapy, acute lymphoblastic leukemia

## Abstract

**Simple Summary:**

Immunotherapy has improved the outcome of relapsed/refractory B-lymphoblastic leukemia. However, little is known about the outcome after recurrence and re-treatment with monoclonal antibodies. The aim of our retrospective study was to evaluate the efficacy and safety of blinatumomab and inotuzumab ozogamicin used for different disease relapses. This multicenter experience of the Campus ALL Italian study group described 71 patients with relapsed/refractory B-lymphoblastic leukemia treated with both blinatumomab and inotuzumab ozogamicin in any sequence. The sequential immunotherapy strategy demonstrated feasibility and efficacy in terms of minimal residual disease, overall and disease-free survival, and as a bridge to allotransplantation.

**Abstract:**

Background: Blinatumomab (Blina) and inotuzumab ozogamicin (InO) has improved the outcome of relapsed/refractory B-lymphoblastic leukemia (R/R B-ALL). However, little is known about the outcome after recurrence and re-treatment with immunotherapy. Methods: We describe 71 R/R B-ALL patients treated for different relapses with Blina and InO. Blina was the first treatment in 57 patients and InO in 14. Twenty-seven patients had a previous allogeneic hematopoietic stem cell transplantation (allo-HSCT). Results: In the Blina/InO group, after Blina, 36 patients (63%) achieved a complete remission (CR), with 42% of negative minimal residual disease (MRD−); after InO, a CR was achieved in 47 patients (82%, 34 MRD−). In the InO/Blina group, after InO, 13 cases (93%) reached a CR (6 MRD−); after Blina, a CR was re-achieved in 6 cases (43%, 3 MRD−). Twenty-six patients proceeded to allo-HSCT. In the Blina/InO group, the median overall survival (OS) was 19 months; the disease-free survival (DFS) after Blina was 7.4 months (11.6 vs. 2.7 months in MRD− vs. MRD+, *p* = 0.03) and after InO, 5.4 months. In the InO/Blina group, the median OS was 9.4 months; the median DFS after InO was 5.1 months and 1.5 months after Blina (8.7 vs. 2.5 months in MRD− vs. MRD+, *p* = 0.02). With a median follow-up of 16.5 months from the start of immunotherapy, 24 patients (34%) are alive and 16 (22%) are alive in CR. Conclusion: In our series of R/R B-ALL, Blina and InO treatment demonstrate efficacy for subsequent relapses in terms of MRD response, OS and DFS, and as a bridge to allo-HSCT.

## 1. Introduction

In patients with acute lymphoblastic leukemia (ALL), classical multi-agent chemotherapy results in complete remission (CR) rates in more than 80% of patients. Nevertheless, despite a high rate of response to induction treatment, the 5-year overall survival (OS) ranges from 75% for young adult patients to 30–40% for patients > 40 years [1,2,3]. Historically, for relapsed/refractory (R/R) ALL occurring in adult patients, survival is unsatisfactory, with long-term survival in <10% [1,2,3]. Novel immunotherapy agents, in particular CAR-T cells and the monoclonal antibodies blinatumomab (Blina) and inotuzumab ozogamicin (InO), have profoundly improved outcomes in R/R B-ALL (B-ALL) patients [4,5,6,7,8,9,10,11,12,13,14,15,16,17,18], compared to traditional chemotherapy, increasing the possibility of performing allogeneic hematopoietic stem cell transplantation (allo-HSCT) with long-term disease control [9,10,11,12,13].

CAR-T cell is the most recent immunotherapy that has become available in this setting, demonstrating high efficacy. A recent large meta-analysis analyzed the results of 38 reports, which enrolled 2134 patients. An overall response rate of 76% was reported, while median OS and event-free survival (EFS) were 36.2 months and 13.3 months, respectively [13].

The study explored many open questions, as key modulators of response, including costimulatory domains, disease status, age, and lymphodepletion. Costimulatory domain 4-1BB in the CAR construct, low-dose cyclophosphamide lymphodepletion, and pretreatment morphologic complete remission seem to be associated with better OS, and morphologic remission and 4-1BB domain are also associated with better EFS. These findings confirm that CAR-T cell therapy may be associated with long-term benefits in R/R B-ALL patients. However, more studies are needed to understand the role of numerous variables on the outcome of CAR-T cell therapy. Among these are the selection criteria of the patients eligible, the best type of construct to use, the type of lymphodepletion regimen, and most importantly, the long-term efficacy and the indication of HSCT consolidation [13]. Furthermore, a direct comparison of CAR-T cells with Blina and InO therapies is lacking. 

Blina is a CD3/CD19-targeting bispecific T-cell engager that recruits CD3^+^ effector T cells to kill CD19^+^ ALL blasts. In the phase III TOWER trial, the rate of CR or CR with incomplete hematologic recovery (CRi) was significantly greater for R/R B-ALL patients treated with Blina compared to patients who received standard-of-care chemotherapy (44% vs. 25%). Of the 271 patients treated with Blina, 24% proceeded to allo-HSCT, with 10% of the patients requiring pre-transplant salvage treatment. After 12 months, the median OS was significantly higher in the Blina group than in the chemotherapy group (8 vs. 4 months) [4]. Blina demonstrated efficacy in monotherapy also in Philadelphia (Ph) positive R/R ALL patients. In the ALCANTARA study [6], 45 patients previously treated with two or more tyrosine kinase inhibitors (TKI) were reported. Sixteen patients obtained CR/CRi after two cycles of Blina. MRD negativity was achieved in 12/14 CR patients. An incidence CRS, neutropenia, and thrombocytopenia of >3 grade occurred in 82% of the patients.

InO is a CD22-directed humanized monoclonal antibody conjugated to calicheamicin. In the phase III INO-VATE trial, in R/R B-ALL patients, InO demonstrated to be superior to the standard of care, with a CR/CRi in 74% versus 31% [5]. Patients in the InO group proceeded to an allo-HSCT in 40% of the cases, without a need for an additional salvage line, compared to 11% of the patients undergoing standard-of-care treatment. The median OS was 7.7 months in the InO group compared to 6.2 months in the standard-of-care group, and the 2-year OS rate was 23% versus 10%, respectively [5].

Nevertheless, although Blina and InO have improved the prognosis of both Ph-negative and Ph-positive R/R B-ALL patients, little is known on the outcome of patients after the recurrence of their disease treatment with the two monoclonal antibodies sequentially. We hereby describe the clinical characteristics and outcome of 71 patients with R/R B-ALL treated with both Blina and InO in any sequence—Blina/InO or InO/Blina—for different disease recurrences, in the context of a multicenter study of the Campus ALL Italian study group.

## 2. Patients and Methods

We conducted a multicenter, retrospective analysis of 71 adult patients with R/R B-ALL treated with Blina and InO in any sequence (Blina/InO or InO/Blina) between March 2013 and February 2020. All procedures were in accordance with the ethical standards of the responsible Ethical Committee on human experimentation (institutional and national) and with the Helsinki Declaration of 1975, as revised in 2013. In each patient, Blina or InO were used for different disease recurrences. Between the two immunotherapies, other treatments could be administered. 

The choice of using Blina or InO first relies on the clinical judgement of the single treating physician.

Blina is approved in Italy for hematologic relapse/resistance and for a blast count of <5% (minimal residual disease, MRD, and persistence), while InO is approved only for hematologic relapse/resistance. Nevertheless, in our study, InO was used in a minority of patients also with a BM blast count of <5%, on the basis of individual assessment and as an off-label prescription.

Patients were grouped according to the antibody that was used first: Blina/InO and InO/Blina groups. Medical records were reviewed to collect demographic, patient-related, disease-related, and clinical outcome data. The expression of CD19 and CD22 were measured by flow cytometry. Response evaluation was performed by a cytological examination of bone marrow (BM) smears. Flow cytometry and polymerase chain reaction (PCR) of leukemia-specific rearrangements of immunoglobulin genes or *BCR::ABL1* transcript levels were used to monitor MRD. CR was defined if a patient had <5% blasts in BM smears and an absence of extramedullary manifestations; a negative MRD response was defined as <0.01% (10^−4^) leukemic cells in the BM. OS was defined as the time from the start of the first immunotherapy agent to death from any cause or last contact, and disease-free survival (DFS) was defined as the time to relapse. 

For the first cycle, Blina was administered by continuous intravenous infusion with a dose of 9 µg/day over 24 h for the first week and 28 µg/day over 24 h for the remaining 3 weeks, followed by a 2-week break. The following cycles were conducted with a dose of 28 µg/day from the beginning. InO was given intravenously at a dose of 1.8 mg/m^2^ for the first cycle and then 1.5 mg/m^2^, divided in three weekly doses. Cycle 1 usually lasted 21 days, and the subsequent cycles were 28 days each. The demographic and disease characteristics are summarized with descriptive statistics. Averaged data were expressed as a median (range). A chi-square test was used to investigate correlations between all the nominal variables described. The Mann–Whitney correlation test was used to compare continuous variables between groups. Survival curves were generated using the Kaplan–Meier method and were compared between groups via the log-rank test. Values of *p* < 0.05 were considered significant; p ns was used to indicate a statistical non-significance. All statistical analyses were performed using STAT VIEW SAS V. 5.0.

An alluvial diagram was elaborated with the RAWGraphs 2.0 software.

## 3. Results

### 3.1. Baselines Characteristics

Seventy-one patients from 19 Italian hematologic centers participating in the Campus ALL network were identified. 

The treatment flow is reported in Figure 1. 

Clinical characteristics are summarized in Table 1.

The median age was 34 years (range 15–64), and the male/female ratio was 1.6 (44/27). Sixteen patients (22%) were Ph-positive B-ALL, 12 in the Blina group (21%) and 4 in the InO group (29%). A t(4; 11) translocation or a complex karyotype were detected in three (4%) and nine (13%) cases, respectively. Most patients (93%) had a ECOG performance status of 0–1. The flow cytometry identified CD19 in all the patients, while CD22 was positive in 62/63 of the tested patients. The CD19 and CD22 median expression levels were 76% and 60%, respectively. The median number of prior therapies was 2 (range 1–9), and the median time between the B-ALL diagnosis and first immunotherapy was 15 months (range 1–179). All the Ph-negative ALL patients received a pediatric-inspired chemotherapy program as the frontline treatment. The Ph-positive ALL patients were treated with intensive chemotherapy plus tyrosine kinase inhibitors (TKIs) in three cases (4%), while 13 patients (18%) were treated with a TKI and steroids in induction without systemic chemotherapy. 

Blina was administered as a first salvage strategy (Blina/InO sequence) in 57 patients (80%) and InO in the other 14 cases (20%) (InO/Blina sequence). Twenty-seven patients (38%) had undergone a previous allo-HSCT. In the Blina/InO population, the median number of previous treatments was 2 (range 1–8); 24/57 (42%) of the patients had received a prior allo-HSCT. For the InO/Blina group, the median number of prior therapies was 3 (range 1–9), and 3/14 (21%) of the patients had undergone an allo-HSCT. No patients received Blina or Ino as part of their frontline treatment. In the Blina/InO group, at the start of Blina, the median white blood count (WBC) was 4.8 × 10^9^/L (range 0.7–98 × 10^9^/L) and the median BM blast count was 40% (range 0–100%). At the start of InO, the WBC was 5.4 × 10^9^/L (range 1.3–101.4 × 10^9^/L) and the median BM blast count was 50% (range 0–90%). In the InO/Blina group, at the start of InO, the WBC was 6.3 × 10^9^/L (range 1–101 × 10^9^/L) and the median BM blast count was 64% (range 2–90%). At the start of Blina, the median white blood count (WBC) was 5.3 × 10^9^/L (range 1.9–72.4 × 10^9^/L) and the median BM blast count was 34% (range 0–90%). Extramedullary involvement at the time of treatment was present in five patients (9%) in the Blina/InO group and in one patient (7%) in the InO/Blina cohort. 

In the Blina/InO group, a median of 2 (range 1–9) Blina cycles and a median of 2 (range 1–6) InO cycles were administered.

In the InO/Blina group, a median of 2 (range 1–4) InO cycles and a median of 1.5 (range 1–4) Blina cycles were administered. 

TKI therapy was associated with immunotherapy in two patients in the Blina/InO group (ponatinib) and in two patients in the InO/Blina group (ponatinib and bosutinib).

Eighteen Blina/InO patients at the time of Blina treatment had <5% bone marrow (BM) blasts (13 pts < 1% BM blasts), and two InO/Blina patients had 3% BM blasts (*p* not significant).

Forty patients received chemotherapy/TKI therapy between their exposures to the two immunotherapy agents (Blina/InO: 33/57, 58%; InO/Blina: 7/14, 50%, *p* not significant). Only two patients received CAR-T cell therapy. The low number of patients treated with CAR-T cells in our series is due to the fact that this immunotherapy was approved in Italy only 6 months before the end of the observational period of the present study. 

The treatments were clofarabine, high-dose cytosine arabinoside, HAM (high-dose cytosine arabinoside and mitoxantrone), high-dose methotrexate, clofarabine plus cyclophosphamide, vincristine, FLAI (fludarabine, high-dose cytosine arabinoside and idarubicin), DLIs (donor lymphocyte infusions), CAR-T cells (two patients), HSCT, L-VAMP (vincristine, methotrexate, cytosine arabinoside, and dexamethasone), Hyper-CVAD, POMP (6-mercaptopurine, vincristine, methotrexate, and prednisone), and ponatinib. 

The sequential therapy institutions and the response to these therapy and interim treatments between the immunotherapies are reported in Figure 1 and Figure 2.

In the Blina/InO group, the median number of treatments between the two immunotherapy agents was 1 (range 0–3), similar to the InO/Blina group (median number of treatments 1, range 0–2) (*p* ns).

### 3.2. Response and outcome

In the Blina/InO group (57 patients), the rate of CR with Blina treatment was 63% (36 patients), including 24/57 patients who achieved a negative MRD (42%); after InO administration, a CR was achieved in 47 patients (82%), with 34/57 (60%) obtaining a negative MRD. In the InO/Blina group (14 patients), after InO, a CR was reached in 13 cases (93%) with a negative MRD in 6 (43%). Blina, used for a subsequent disease recurrence, allowed for the achievement of CR in 6 patients (43%), with 3/14 (21%) patients achieving a negative MRD.

We also investigated the impact of leukemia bulk on the response during the Blina and InO therapies in the Blina/InO and InO/Blina groups and found that blasts of >50% were associated with a significantly worse CR rate after the Blina therapy, while no influence of the blast percentage was evidenced after the InO treatment. 

In particular, in the Blina/InO group, CR was obtained in 45% of the patients with BM blasts of <50% vs. 13% of those with BM blasts of >50% (*p* = 0.04).

Similar results were obtained in the InO/Blina group after the Blina treatment (CR of 46% with BM blasts of <50% vs. 8% with BM blasts of >50%, *p* = 0.05).

No statistically significant differences in terms of CR were observed for the Ph+ and Ph-negative patients in both the Blina/InO and InO/Blina groups.

Immunotherapy was used as a bridge to allo-HSCT in 26 patients (37%) of whom 24 were in the Blina/InO population and 2 were in the InO/Blina group, and 14 of these were represented by a second HSCT.

Donor lymphocyte infusions were performed in eight cases (11%).

The median OS from the first immunotherapy was longer in the Blina/InO cohort than in the InO/Blina group, but statistical significance was not reached (21.9 months vs. 15.3 months, respectively, *p* = ns, Figure 3). 

From the first immunotherapy, in the Blina/InO group, the median OS was 19 months and after InO, 6.3 months. The OS in the patients who reached an MRD negativity was not significantly different compared to that of the patients who remained MRD+. DFS after Blina was 7.4 months and was significantly better in MRD− compared to MRD+ patients (11.6 vs. 2.7 months, *p* = 0.03). After InO, DFS was 5.4 months, with no significant difference between the patients who became MRD− or did not. In the InO/Blina group, the median OS was 9.4 months and after Blina, 4.6 months, and it was significantly better in patients who obtained an MRD negativity (7.5 vs. 2.8 months, *p* = 0.02). In the InO/Blina group, the median DFS was 6.6 months (5.1 after the start of InO and 1.5 months after Blina), and it appears to be longer in patients who became MRD− after Blina (*p* = 0.02). Overall, the patients who reached negative MRD statues after the second-line immunotherapy witnessed a significantly better OS (*p* = 0.001). Interestingly, the OS and DFS from the first immunotherapy were longer in the group of patients who had undergone an allo-HSCT (data available for 64/71 patients) prior to starting the immunotherapy: the median OS was 24.2 vs. 13 months (*p* = 0.022). Allo-HSCT after the second immunotherapy was associated with a better OS and DFS, which was not significant (median OS, 9.8 months and median DFS, 7.2 months vs. 7.8 and 4.4 months).

With a median follow-up time of 16.5 months from the start of the immunotherapy and 33.8 months from the initial diagnosis, 24 patients (34%) are still alive and 16 (22%) are alive in CR. In the Blina/InO group, 24% of the patients are alive and in CR, while in the InO/Blina cohort, three patients (21%) are still alive and in CR.

No statistically significant differences in terms of DFS and OS were observed for the Ph-negative and Ph+ patients in both the Blina/InO and InO/Blina groups.

The cause of death was attributed to ALL progression in 34 patients (48%), infection in 6 patients (8%), allo-HSCT-related complications in 6 (8%), and bleeding complications in 1 case. Five patients (7%) died in CR due to a veno-occlusive disease during allo-HSCT after InO salvage treatment. Even if the study was not designed to evaluate the incidence of VOD and its possible correlation with a cumulative InO dose, in our series, the patients who developed a fatal VOD presented a total dose of InO (median number of cycles 2, range 1–2) comparable to the other patients who received allo-HSCT (median number of cycles 2, range 1–4) without the occurrence of this fatal complication.

### 3.3. Toxicity

In the Blina/InO group, the Blina treatment was complicated by grade > 3 adverse events (AEs) in 15 cases (26%); 3 patients (20%) had hematologic toxicities, while the remaining 12 (80%) experienced an extra-hematologic AE, represented by neurotoxicity in 4 (27%). Infectious complications occurred in 17 patients (31%). 

The InO treatment was complicated by grade > 3 adverse events (AEs) in 12 cases (21%); nine patients (16%) had hematologic toxicities, while the remaining three (0.5%) experienced extra-hematologic AEs, which were one CMV enterocolitis, one sacroiliitis, and one hepatic toxicity. Infectious complications occurred in 21 patients (37%).

For the InO/Blina group, exposure to InO was associated with grade > 3 AEs in three patients (21%), with one case (7%) of hematologic toxicity and two cases (14%) of extra-hematologic AEs (one liver function impairment). Infectious complications occurred in four cases (28%). The Blina treatment was associated with grade > 3 AEs in three patients (21%), with one case (7%) of hematologic toxicity and two cases (14%) of extra-hematologic AEs (one pulmonary thromboembolism and one deep vein thrombosis). Infectious complications occurred in four cases (28%).

## 4. Discussion

The landscape of the therapeutic approach to R/R B-ALL has rapidly evolved in recent times, given the availability of potent immunotherapies such as Blina, InO, and more recently, CAR-T cells.

Furthermore, these agents, in particular Blina and InO, are more and more used in association and are sequenced with chemotherapy and TKIs, with encouraging results in terms of an increasing OS and DFS [12].

In this scenario, CAR-T cell therapy was recently introduced in the therapeutic armamentarium of R/R B-ALL, with very promising outcomes [13,19,20,21]. 

In particular, CAR-T cell therapy was approved in Italy in August 2019, a few months before the end of the enrollment period of our study, and therefore, it did not have a significant impact on the treatment choice of our series of patients.

Nevertheless, with its wider use in the immediate future, several clinical questions will emerge, among which concern the effective long-term benefit, the best CAR-T construct and cellular source, the most efficacious lymphodepletion regimens, and the best sequence with HSCT and their use in first line [13,19,20,21]. Following the expansion of their application, appropriate guidelines will be needed in order to assign patients to the best treatment choice and timing among Blina, InO, and CAR-T cells. 

Immunotherapy with Blina and InO has profoundly changed the outcomes of both R/R Ph-negative and Ph+ B-ALL, improving significantly the prognosis of adult patients [4,5,6,7,8,9,10,11,12,13,14,15,16,17,18], and new strategies for their use in maintenance, MRD eradication, and debulking prior to Blina and the association of different combinations/sequencing of Blina and InO with a reduced intensity chemotherapy are being explored.

In particular, Blina maintenance was investigated in a randomized, phase 3 study of Blina vs. SOC in adults with R/R Ph-negative B-ALL, and this demonstrated that patients achieving remission with Blina had a longer OS with maintenance Blina therapy [22]. Moreover, both Blina and InO have been used in MRD-positive patients: Blina has demonstrated efficacy in eradicating MRD in Ph-negative B-ALL in the BLAST study [15], and InO is currently being investigated in this setting both in Ph-negative and Ph+ B-ALL patients who are candidates of HSCT (GIMEMA ALL2418, NCT03610438) [8].

As a consequence of the promising results observed, these studies have prompted the exploration of further innovative approaches with Blina and InO in association and/or sequencing in the attempt to enhance their efficacy.

From this perspective, as tumor bulk has been demonstrated to have a significant impact on Blina therapy outcomes, an ongoing trial is investigating the role of InO to decrease disease burdens, followed by Blina to maintain remission (NCT03739814) in B-ALL R/R patients or in newly diagnosed B-ALL elderly patients. This approach is particularly interesting for InO’s ability to induce deep and rapid responses, potentially optimizing the subsequent Blina treatment and at the same time, appropriately distancing HSCT from InO treatment, reducing, in this way, the InO risk of VOD.

As a last point, the association of InO, with or without Blina, with a reduced intensity chemotherapy (mini-Hyper-CVD) was successfully used as salvage therapy in Ph-negative B-ALL patients [11], with a significant enhancement in efficacy, maintaining a manageable toxicity, especially after reducing the chemotherapy intensity. With this approach, younger patients had overall and complete responses of 83% and 63%, with a high MRD negativity (82% of responders). Hepatic sinusoidal obstruction syndrome decreased significantly from 13% to 2% after reducing the regimen intensity. With a median follow up of 48 months, the median OS was 17 months and at 3 years, the OS was 34% with mini-Hyper-CVD plus InO and 52% with additional Blina. Interestingly, HSCT did not improve the OS [11], suggesting that its role after the introduction of these combined treatments should be further investigated and may change in the future.

Blina and InO have therefore become the standard of care for the treatment of R/R, and Blina has also become the standard of care for the eradication of persistent MRD. 

These studies anticipated the introduction of these agents also in the first-line treatment of Ph+ and Ph-negative B-ALL patients in different combinations and sequences with a TKI and/or chemotherapy.

In the Ph+ ALL patients, the most interesting strategy is represented by the attempt to use a chemo-free induction regimen approach, adding a TKI to the Blina treatment. In particular, Blina plus dasatinib or ponatinib used in the front line, demonstrated optimal outcomes in terms of response rates, OS, and DFS [23,24,25]. In addition, this approach is currently being investigated in a randomized phase 3 study that compares upfront Blina plus ponatinib treatment versus the standard regimen, represented by intensive chemotherapy plus imatinib (GIMEMA ALL2820, NCT04722848).

On the other hand, in the Ph-negative patients, Blina has been added in the first line to standard chemotherapy (GIMEMA LAL2317, NCT03367299) [26] and has also been associated with InO and modified mini-Hyper-CVD [27], in this case also demonstrating manageable toxicity and high efficacy.

Lastly, Blina has also been introduced as a consolidation regimen after first line treatment, reporting promising results [28].

Nevertheless, few data are available on the prognosis of R/R B-ALL salvaged with Blina or InO after a previous first line of immunotherapy.

In our multicenter, retrospective Campus ALL study, we evaluated R/R B-ALL patients in which Blina and InO were used for different disease recurrences. There are few published analyses on adult R/R B-ALL patients who received both Blina and InO treatments. Recently, Badar et al. reported a subset analysis on 61 out of a total of 276 patients treated with immunotherapy, who received Blina and InO for different relapses [29].

The sequence of Blina and InO was administered to 57 cases, while the InO and Blina sequence was used in the remaining 14 patients. The baseline characteristics were comparable. Forty patients received very etherogeneous additional treatments between the two immunotherapies. Even if this may have introduced possible biases in the outcomes, nevertheless, the global frequency of all these treatments were comparable in the two groups.

The median number of cycles for the second agent was 1 in the InO group (range 1–6) and 1 in the Blina group (range 1–4). The CR/CRi rate was 58% or 52% for the patients who received InO or Blina, respectively, as the second immunotherapeutic drug.

CR is significantly better in patients starting the Blina treatment with <50% BM blasts, confirming the reported relevance of debulking treatment prior to Blina delivery, in order to optimize its efficacy, while no impact of the blast number seems to be evident for the InO therapy [30,31].

Historically, the response rate in patients with R/R B-ALL has been reported to be inferior in subsequent lines of chemotherapy, from approximately 40% in first salvage to 20% after the second/third salvage regimens [1,2,3]. Differently from conventional therapy, we have observed higher CR rates, similar to those reported in the seminal trials using Blina and InO in the setting of R/R B-ALL patients [4,5]. Similar to Badar’s study, more than half of our patients relapsed or progressed after the first-line immunotherapy was salvaged with the use of the second immunotherapeutic agent [29]. In our Blina/InO group, the rate of CR after Blina was 63%, and after InO, it increased to 82%, with 60% of the patients obtaining a negative MRD. The InO/Blina group was also characterized by a high CR rate and a significant possibility of reaching MRD negativity, also after the second immunotherapy. As in the analysis by Badar et al., the median OS appeared to be longer, even if not significantly, for patients who received InO as the second immunotherapeutic agent [29].

The TOWER and INO-VATE clinical trials also suggested that Blina and InO are successful salvage options for patients with R/R B-ALL and bridge them to an allo-HSCT to obtain a long-term cure [4,5], and recent studies suggest that a combined approach will be the winning strategy to optimize the response to these drugs.

In our report, approximately 40% of the patients were successfully bridged to allo-HSCT and had the possibility of taking advantage of the described benefit of allo-HSCT after immunotherapy in improving the duration of the response and OS [9,12]. In our study, allo-HSCT was associated with a longer survival duration, albeit not reaching statistical significance.

Interestingly, we observed that OS and DFS from the first immunotherapy were superior in patients who had relapsed after a previous allo-HSCT, possibly due to a role of CD3 lymphocye of the donor, which may be more efficient as immunologic effectors once engaged by Blina.

When selecting immunotherapy (Blina or InO), potential AEs are one of the most important considerations in order to select the optimal treatment, considering both patient-related and disease-specific needs. The toxicity profile of Blina mainly consists of the cytokine release syndrome and of neurologic symptoms, while InO targets the liver with sinusoidal obstruction syndrome, in particular after HSCT [4,5,6,7,8,10,11]. In our study, about 25% of the patients experienced ≥ G3 AEs, not significantly different in the Blina/InO and InO/Blina groups and comparable to those observed in the TOWER and INO-VATE trials [4,5]. The cause of death was mainly attributed to ALL’s progression, infection, and allo-HSCT-related complications (8%, mainly due to veno-occlusive disease). 

## 5. Conclusions

The approval of novel immunotherapeutic agents has markedly impacted the outcome of R/R B-ALL patients. The treatment choice requires that multiple factors are considered, including both patient-related and disease-specific characteristics.

In the present study, we described a very difficult clinical scenario, regarding those patients experiencing subsequent relapses and those who have been treated with both Blina and InO in any sequence, also with intercurrent salvage therapies between the two immunotherapies. This excludes all the patients who were cured after treatment with only Blina or InO.

Nevertheless, the clinical outcomes of these patients are surprisingly similar to those reported in patients receiving only either one of the two agents, underlining the favorable clinical impact of Blina and InO applied even in what may be considered the worst scenario for R/R B-ALL patients, giving a strong rationale for their sequential use also in those patients who obtain a profound CR after the application of the first one. One possibility would be to use inotuzumab as the first agent in order to obtain an efficient debulking/CR and blinatumomab in the consolidation phase, with two rationales: distancing the InO treatment before eventual HSCT, therefore reducing the described risk of VOD and favoring patients’ immune reconstitution, possibly improving the efficiency of CD3 lymphocyte immunological effectors engaged by Blina.

Our real-life multicenter study on R/R B-lineage ALL patients with multiple previous lines of treatment demonstrates the feasibility and efficacy of a sequential immunotherapy strategy in terms of MRD response, DFS, and OS, and as an effective bridge to allo-HSCT to treat subsequent relapses. Future studies are required to determine how to best combine and sequence these agents to achieve the best outcomes in R/R B-lineage ALL patients in order to reduce the incidence of relapse.

## Figures and Tables

**Figure 1 cancers-15-04623-f001:**
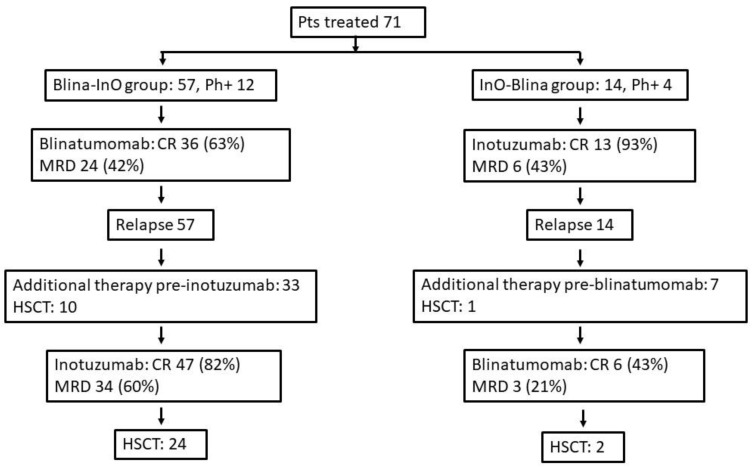
Diagram of the treatment flow of the Blina/InO and InO/Blina groups.

**Figure 2 cancers-15-04623-f002:**
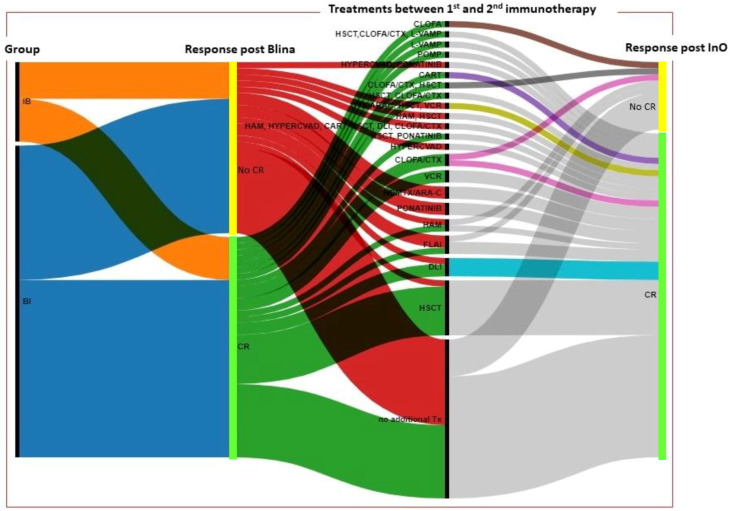
This figure reports the response after InO or after Blina in the IB and BI groups: CR patients cluster in the light green bars, and no CR patients cluster in the yellow bars. In particular, for the response post InO, the patients that obtained a CR (clustering in the light green bar) are represented in light grey, cyan, pink (1 patient), violet (1 patient), and dijon (1 patient) and those who did not obtain CR (clustering in the yellow bars) are represented in light grey, dark grey (1 patient), pink (1 patient), and brown (1 patient). IB: InO/Blina group, BI: Blina/InO group, CR: Complete remission, CLOFA: clofarabine, CTX: cyclophosphamide, Tx: therapy.

**Figure 3 cancers-15-04623-f003:**
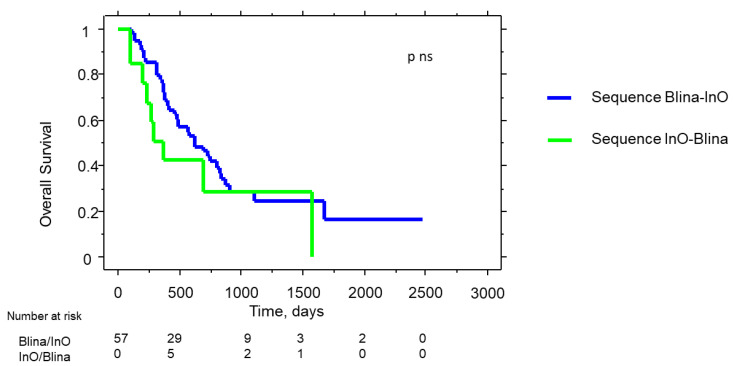
Overall survival in Blina/InO versus InO/Blina patients from the first immunotherapy. Blina/InO = blinatumomab and subsequent inotuzumab-ozogamicin; InO/Blina = inotuzumab-ozogamicin and subsequent blinatumomab.

**Table 1 cancers-15-04623-t001:** Clinical characteristics and toxicities of Blina/InO and InO/Blina groups at the start of the first and second immunotherapy. ECOG was reported at the start of the first immunotherapy. No significant differences between groups, except for age (* *p* = 0.001).

Blina/InO Group	No. 57	InO/Blina Group	No. 14
**Age—median (range) pre Blina**	33 (15–64) *	**Age—median (range) pre InO**	41.5 (22–64) *
**Male/female**	34/23	**Male/female**	10/4
**ECOG n° (%) pre Blina**		**ECOG n° (%) pre InO**	
0	33	0	8
1	21	1	4
2	2	2	2
**Treatments pre-Blina—median (range)**	2 (1–8)	**Treatments pre-InO—median (range)**	3 (1–9)
**Previous HSCT—n° (%)**	24/57 (42%)	**Previous HSCT—n° (%)**	3/14 (21%)
**WBC (×10^9^/L)—median (range) pre Blina**	4.8 (0.7–98)	**WBC (×10^9^/L)—median (range) pre InO**	6.3 (1–101)
**WBC (×10^9^/L)—median (range) pre InO**	5.4 (1.3–101.4)	**WBC (×10^9^/L)—median (range) pre Blina**	5.3 (34)
**Bone marrow blast median % (range) pre Blina**	40 (0–100)	**Bone marrow blast median % (range) pre Ino**	64 (2–90)
**Bone marrow blast median % (range) pre InO**	50 (0–90)	**Bone marrow blast median % (range) pre Blina**	34 (0–90)
**Ph+—n° (%)**	12/57 (21%)	**Ph+—n° (%)**	4/14 (29%)
**Extramedullary involvement—n° (%) pre Blina**	5/57 (9%)	**Extramedullary involvement—n° (%) pre InO**	1/14 (7%)
**Blina cycles—median (range)** **InO cycles—median (range)**	2 (1–9) 2 (1–6)	**InO cycles—median (range)** **Blina cycles—median (range)**	2 (1–4) 1.5 (1–4)
**Toxicity G3/4—n° (%) after Blina**	15/57 (26%)	**Toxicity G3/4—n° (%) after InO**	3/14 (21%)
Hematological	3	Hematological	1
Extrahematological	12	Extrahematological	2
Neurological	4	Hepatic	1
**Infectious AEs—n° (%)**	17/57 (31%)	**Infectious AEs—n° (%) after InO**	4/14 (28%)
**Toxicity G3/4—n° (%) after InO**	12/57 (21%)	**Toxicity G3/4—n° (%) after Blina**	3/14 (21%)
Hematological	9	Hematological	1
Extrahematological	3	Extrahematological	2
1 CMV enterocolitis, 1 sacroiliitis, 1 hepatic		Pulmonary thromboembolism, deep vein thrombosis	
**Infectious AEs—n° (%) after InO**	21/57 (37%)	**Infectious AEs—n° (%) after Blina**	4/14 (28%)

## Data Availability

The original contributions presented in this study are included in the article, and further inquiries can be directed to the corresponding author.

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
