# Peer review of "Blinatumomab and Inotuzumab Ozogamicin Sequential Use for the Treatment of Relapsed/Refractory Acute Lymphoblastic Leukemia: A Real-Life Campus All Study"

_cancers, 2023, doi:10.3390/cancers15184623_

Round 1

Reviewer 1 Report

Fracchiolla and colleagues from Italy present a retrospective study on sequencing blinatumomab and inotuzumab in r/r ALL. The paper is well written and clear, but the concept is not entirely novel, and this was presented before by Badar T et al. Cancer 2021.  Nonetheless, it is a valuable analysis support the concept that the use of inotuzumab and blinatumomab in relapsed/refractory B-ALL is complementary, safe and effective.

·        The manuscript will benefit from a consort diagram illustrating how many patients treated, how many subsequently responded, how many transplanted in CR, and then how many relapsed and subsequently treated with the other agent.  

·        Can authors confirm that all patients included in this analysis had marrow blasts 5% and no patient had MRD only disease?

·         Authors should state which TKI used concurrently with blina/InO in Ph+ ALL patients.

·         How many of post response transplants were second transplant?

·      For extramedullary involvement, do authors refer to the time of diagnosis or the time of treatment with blina/InO?

It is acceptable. 

Author Response

Answers to Comments and Suggestions Reviewer 1

We  thank the Reviewer for the observations to which we have answered:

Reviewer comment:

Fracchiolla and colleagues from Italy present a retrospective study on sequencing blinatumomab and inotuzumab in r/r ALL. The paper is well written and clear, but the concept is not entirely novel, and this was presented before by Badar T et al. Cancer 2021.  Nonetheless, it is a valuable analysis support the concept that the use of inotuzumab and blinatumomab in relapsed/refractory B-ALL is complementary, safe and effective.

Question:·        The manuscript will benefit from a consort diagram illustrating how many patients treated, how many subsequently responded, how many transplanted in CR, and then how many relapsed and subsequently treated with the other agent.  

Answer: We have created a consort diagram as requested

Q:·        Can authors confirm that all patients included in this analysis had marrow blasts ≥5% and no patient had MRD only disease?

A: we have reported in the Results the data requested:

“Eighteen Blina/InO patients presented at the time of Blina treatment <5% bone marrow (BM) blasts (13 pts <1% BM blasts), and 2 InO/Blina patients 3% BM blasts. The presence of patients in MRD with <1%BM blasts in Blina/InO group, before first immunotherapy, is due to the availability of Blina in this setting, differently from InO (p not significant).

Q:·         Authors should state which TKI used concurrently with blina/InO in Ph+ ALL patients.

A: we have reported in the Results the data requested:

“Two patients in the Blina/InO group (ponatinib) and 2 in the InO/Blina group received immunotherapy with a TKI (ponatinib and bosutinib).”

Q:·         How many of post response transplants were second transplant?

“(…) and 14 of these were represented by second HSCT.”

Q:·      For extramedullary involvement, do authors refer to the time of diagnosis or the time of treatment with Blina/InO?

 A: we have reported in the Results the data requested:

“Extramedullary involvement at the time of treatment (…)”

Reviewer 2 Report

Fracchiolla and others describe the off-study, real-world experience across the Campus ALL network using blinatumomab (blina) and inotuzumab ozogamicin (InO) for adults with relapsed/refractory B-cell ALL (R/R B-ALL). The optimal use and sequence of these drugs is not yet known, and analyses like this could help close this knowledge gap. Unfortunately, based on the way that patients were selected for this analysis, a VERY important subgroup of patients were excluded: As I understand their design, patients who received either of these immunotherapy agents and then achieved durable remission as a result of this treatment are not included here. If someone receives Blina or InO and is cured, they will never need the other agent. If a relatively large number of such patients were excluded from this analysis, this would significantly underestimate the impact that these agents has on the management of R/R B-ALL. As currently designed, the authors have only looked at the worst outcomes possible (i.e., those for whom the first drug failed). This and other aspects of their design and description introduce bias that the authors should try to clarify more thoroughly.

Major Comments:

1.       Introduction, page 2 (paragraph 1): This cohort of patients include both Ph+ and Ph- disease. However, the background data focus solely on Ph- disease for blinatumomab. Please include some description of the results of the Phase II ALCANTARA trial of blinatumomab for Ph+ ALL (Martinelli and others, J Clin Oncol, 2017;35:1795-1802).

2.       Patients and Methods, page 3 (paragraph 2): Please provide a description of how these agents were chosen. Was it strictly at the discretion of the individual treating physicians? Were there any specific network or national regulations in place that limited the use of either drug to specific populations?

3.       Results, page 3: Please provide a Table summarizing the key baseline characteristics and how they compare between the two study groups.

4.       Results, page 3: Unless I missed it, please describe how many patients in each group received other treatments between the two immunotherapy agents. In the Methods, it is stated how “two immunotherapies could be sequential or interspersed with other treatments.” The details of this could potentially identify another source of bias.

5.       Discussion, page 6: Please provide a description of some of the limitations of this study. If no significant changes to the eligibility criteria for this analysis are pursued, please address the comment above about how only patients who experienced failure of the first immunotherapy agent were included. Please also address some of the sources of bias referenced above.

Minor Comments:

1.       Introduction, page 2 (line 79): Blinatumomab is technically a bispecific T-cell engager, not a “bispecific monoclonal antibody.” Please revise this.

2.       Patients and Methods, page 3 (line 113) and elsewhere: The preferred nomenclature for gene fusions is now a double-colon. Thus, please change “BCR-ABL” to “BCR::ABL1” throughout.

3.       Results, page 4 (lines 151-152): I do not understand this statement, “No prior therapies with Blina or InO were administered.” Does this mean that no patients received these agents as part of their frontline treatment? Please reword this sentence to clarify.

As a native English speaker, I felt this was well-written with only some minor issues noted.

Author Response

Answers to Comments and Suggestions Reviewer 2

 We  thank the Reviewer for the observations to which we have answered:

Reviewer comment:

Fracchiolla and others describe the off-study, real-world experience across the Campus ALL network using blinatumomab (blina) and inotuzumab ozogamicin (InO) for adults with relapsed/refractory B-cell ALL (R/R B-ALL). The optimal use and sequence of these drugs is not yet known, and analyses like this could help close this knowledge gap. Unfortunately, based on the way that patients were selected for this analysis, a VERY important subgroup of patients were excluded: As I understand their design, patients who received either of these immunotherapy agents and then achieved durable remission as a result of this treatment are not included here. If someone receives Blina or InO and is cured, they will never need the other agent. If a relatively large number of such patients were excluded from this analysis, this would significantly underestimate the impact that these agents has on the management of R/R B-ALL. As currently designed, the authors have only looked at the worst outcomes possible (i.e., those for whom the first drug failed). This and other aspects of their design and description introduce bias that the authors should try to clarify more thoroughly.

Major Comments:

  1. Question: Introduction, page 2 (paragraph 1): This cohort of patients include both Ph+ and Ph- disease. However, the background data focus solely on Ph- disease for blinatumomab. Please include some description of the results of the Phase II ALCANTARA trial of blinatumomab for Ph+ ALL (Martinelli and others, J Clin Oncol, 2017;35:1795-1802).

Asnswer: We have included a short description of ALCANTARA study in the Introduction section.

  1. Q: Patients and Methods, page 3 (paragraph 2): Please provide a description of how these agents were chosen. Was it strictly at the discretion of the individual treating physicians?

Were there any specific network or national regulations in place that limited the use of either drug to specific populations?

A: We introduced in the Patients and Methods a short description of how Blina or InO has been chosen and of national limitations to the use of the two drugs

“The choice of using Blina or InO first, rely to clinical judgement of the single treat-ing physician. Blina was available in the case of blast <5%, while InO only for hematologic relapse /resistance.”

  1. Q: Results, page 3: Please provide a Table summarizing the key baseline characteristics and how they compare between the two study groups.

A: A table has been included, as requested

  1. Q: Results, page 3: Unless I missed it, please describe how many patients in each group received other treatments between the two immunotherapy agents. In the Methods, it is stated how “two immunotherapies could be sequential or interspersed with other treatments.” The details of this could potentially identify another source of bias.

A: We reported in the Results the data requested and discussed it in the Discussion as a potential bias source:

Results: “Forty pts received intercurrent chemotherapy/TKI therapy, (33/57 Blina/InO, 58%, 7/14 InO/Blina, 50%, p NS): clofarabine, high dose cytosine arabinoside, HAM (high dose cytosine arabinoside and mitoxantrone), high dose methotrexate, clofarabine-cyclophosphamide, vincristine, FLAI (fludarabine, high dose cytosine arabinoside, idarubicin), donor lymphocyte infusions, CART, HSCT, L-VAMP (vincristine, methotrexate, cytosine arabinoside, dexamethasone), Hyper-CVAD, POMP (6-mercaptopurine, vincristine, methotrexate, prednisone), ponatinib. More than one treatment is possible for single patient.”.

Discussion: “Forty patients received very etherogeneous additional treatments between the two immunotherapies (58% in the Blino/InO and 50% in the InO/Blina group, p NS). Even if this may  have introduced possible biases in the outcomes, nevertheless the global frequency of all these treatments were comparable in the two groups.”

  1. Q: Discussion, page 6: Please provide a description of some of the limitations of this study. If no significant changes to the eligibility criteria for this analysis are pursued, please address the comment above about how only patients who experienced failure of the first immunotherapy agent were included. Please also address some of the sources of bias referenced above.

A: We introduced the required comments

“In the present study we describe a very difficult clinical scenario, regarding those patients experiencing subsequent relapses, and who have been treated with both Blina and InO in any sequence, also with intercurrent salvage therapies between the two immunotherapies. excluding all the patients who have been cured after treatment with only Blina or InO.

Nevertheless, the clinical outcomes of these patients are surprisingly similar to those reported in patients receiving only either one of the two agents, underlining the favorable clinical impact of Blina and InO applied even in this that may be considered the worst scenario for RR ALL patients, giving a strong rationale for their sequential use also in those patients who obtain a profound CR after the application of the first one. One possibility would be to use inotuzumab as first agent, in order to obtain an ef-ficient debulking/CR, and blinatumomab in consolidation phase, with two rationales: distancing InO treatment before eventual HSCT, therefore reducing the described risk of VOD, and favoring patients’ immune reconstitution, possibly improving the effi-ciency of CD3 lymphocytes immunological effectors engaged by Blina.”

Minor Comments:

  1. Q: Introduction, page 2 (line 79): Blinatumomab is technically a bispecific T-cell engager, not a “bispecific monoclonal antibody.” Please revise this.

A: we revised as requested

  1. Q: Patients and Methods, page 3 (line 113) and elsewhere: The preferred nomenclature for gene fusions is now a double-colon. Thus, please change “BCR-ABL” to “BCR::ABL1” throughout.

A: we revised as requested

  1. Q: Results, page 4 (lines 151-152): I do not understand this statement, “No prior therapies with Blina or InO were administered.” Does this mean that no patients received these agents as part of their frontline treatment? Please reword this sentence to clarify.

A: we clarified the sentence as requested

Reviewer 3 Report

Fracchiolla and colleagues present here an important study on the outcomes of patients with relapsed/refractory (R/R) B-cell ALL based on the sequence of use of blinatumomab and inotuzumab. As these drugs are both now standard of care in R/R ALL, it is important to understand their sequential effectivity for proper clinical decision making. This makes the data very relevant. The numbers were small (especially in the InO/Blina group) to make any intergroup comparisons. Important points that need to be further addressed are as follows:

Major points:

1) The disease burden at which these immunotherapies are introduced have  possible impact on subsequent remission rates and duration. Did all patients have morphological disease when these drugs were initiated in both groups and at both time points or were they initiated also for persistent/recurrent MRD positivity? This needs to be highlighted and discussed clearly. 

2) In the "Methods" section, it is mentioned that Blina was available only for blasts <5%, however in Table 1, we do see under the Blina-Ino group that median BM blast at Blina initiation was 40%. This statements are contradictory and need to be clarified. 

3) There were heterogeneous salvage chemotherapy instituted between the Ino --> Blina or Blina --> InO. This makes the true assessment of the effectivity of these drugs difficult. One good way would be to draw an Alluvial plot and have the sequential therapy institutions, response to these therapy and interim chemotherapy treatment between the immunotherapies as 3 possible timepoints (however the authors could chose the best way they want to depict the data).

4) Were any of these patient treated or considered for therapy with anti CD-19 standard of care CAR T-cell therapy. As CAR T is another approved option in R/R B-cell ALL, the applicability, use and outcomes with such therapy in these situations need to be described in the results and discussed.

5) Though InO and Blina are approved as monotherapy, data from both the INO-VATE and TOWER trial showed that responses with these therapies are short lived. Indeed the maximum response with these drugs seem to be from a combination approach as has been shown from multiple publications from MDACC (Kantarjian et al, JHO, May 2023). This need to be addressed more in the "Discussion" section.

6) Hepatic SOS/VOD is a know adverse event with InO and negatively impacted by dose of InO, SCT etc. In the present study 7 patients died in CR due to SOS. What was the overall incidence of SOS? Additionally what were the cumulative doses of InO in both the arms? Was there any difference in the cumulative dose of the patients who developed and did not develop SOS after InO. This could be presented in more details.

Minor points: 

1) In Table 1, the header for the second group should be Ino/Blina. As of now it says only "InO"

2)In line 247 the statement lacks the word after "profoundly changed"; I presume the authors meant "outcomes"

3) Was there any difference in the outcomes of Ph+ and Ph- patients in both groups combined? 

4) The title could contain the terms "Sequential use"

5) Is there a Figure 1? It seems the first figure is "Figure 2"? Additionally, consider adding "Numbers at risk" below the x-axis of Figure 2. 

Acceptable level of English 

Author Response

We thank the reviewer for the observations, to which we have tried to answer on a point by point basis.

Major points:

1. The disease burden at which these immunotherapies are introduced have  possible impact on subsequent remission rates and duration. Did all patients have morphological disease when these drugs were initiated in both groups and at both time points or were they initiated also for persistent/recurrent MRD positivity? This needs to be highlighted and discussed clearly. 

Answer: following Reviewer's request, we reported the percentage of BM blasts in both groups and at both time points. No statistical differences were observed in the frequencies, while we evidenced in patients treated with blinatumomab in both groups a better rate of CR in patients with <50% blasts. This was not observed for InO therapy.

We reported these data in the Results and in the Discussion sections, and we highlighted the importance of debulking prior of blinatumomab treatment.

2. In the Methods section, it is mentioned that Blina was available only for blasts <5%, however in Table 1, we do see under the Blina-Ino group that median BM blast at Blina initiation was 40%. This statements are contradictory and need to be clarified. 

Answer: we apologize for the misunderstanding. We have clarifyed the point in the text.

In summary, drug approval for blinatumomab in Italy is for hematologic relapse and MRD persistence, while for InO is only for hematologic relapse. Nevertheless, for some patients, InO has been used, in an off label prescription, also in some cases with bone marrow blast <5%, and we have better specified these aspects in the text.

3. There were heterogeneous salvage chemotherapy instituted between the Ino --> Blina or Blina --> InO. This makes the true assessment of the effectivity of these drugs difficult. One good way would be to draw an Alluvial plot and have the sequential therapy institutions, response to these therapy and interim chemotherapy treatment between the immunotherapies as 3 possible timepoints (however the authors could chose the best way they want to depict the data).

We have produced the alluvial plot requested (Fig. 2)

4. Were any of these patient treated or considered for therapy with anti CD-19 standard of care CAR T-cell therapy. As CAR T is another approved option in R/R B-cell ALL, the applicability, use and outcomes with such therapy in these situations need to be described in the results and discussed.

Answer: Our study has been designed and completed for the large part before CART cellular therapy was available in Italy.

We have highlighted this point in the text and addressed briefly the outcomes of CART therapy in R/R B ALL and added a comment on the open clinical issues in the choice among immunotherapies available at the present moment, with relative references.

5. Though InO and Blina are approved as monotherapy, data from both the INO-VATE and TOWER trial showed that responses with these therapies are short lived. Indeed the maximum response with these drugs seem to be from a combination approach as has been shown from multiple publications from MDACC (Kantarjian et al, JHO, May 2023). This need to be addressed more in the "Discussion" section.

Answer: We have reported in the Discussion section a number of studies using the cited associations

6. Hepatic SOS/VOD is a know adverse event with InO and negatively impacted by dose of InO, SCT etc. In the present study 7 patients died in CR due to SOS. What was the overall incidence of SOS? Additionally what were the cumulative doses of InO in both the arms? Was there any difference in the cumulative dose of the patients who developed and did not develop SOS after InO. This could be presented in more details.

Answer: We have addressed these concerns reporting the data requested in the Result section, as follows:

“Even if the study was not designed to evaluate the incidence of VOD and its possible correlation with cumulative InO dose, in our series the patients who developed a fatal VOD presented a total dose of InO (median number of cycles 2, range 1-2) comparable to the other patients who received allo-HSCT (median number of cycles 2, range 1-4) without developing this fatal complication.”

Minor points: 

1. In Table 1, the header for the second group should be Ino/Blina. As of now it says only "InO"

Answer: We corrected as requested

2. In line 247 the statement lacks the word after "profoundly changed"; I presume the authors meant "outcomes"

Answer: We corrected as requested

3. Was there any difference in the outcomes of Ph+ and Ph- patients in both groups combined?

Answer: No statistically significant differences in terms of CR, DFS AND OS were observed for Ph pos and Ph neg patients in both Blina/InO and InO/Blina groups.

We have reported in the Results section these data

4. The title could contain the terms "Sequential use"

Answer: We corrected as requested

5. Is there a Figure 1? It seems the first figure is "Figure 2"? Additionally, consider adding "Numbers at risk" below the x-axis of Figure 2. 

Answer: The Fig 1 was represented by a consort diagram of the treatment flow of the patients.

We have embedded it in the text following the reviewer observation

Round 2

Reviewer 2 Report

Thank you for addressing my concerns. Based on these responses, it would appear that the authors have opted to maintain this manuscript as a description of the outcomes of patients treated with both blina and InO, as opposed to trying to explore the importance of sequencing these agents. This is perfectly acceptable.

Comments:

1.       The addition of the data from ALCANTARA is welcome, but it seems out of place where it was added to the Introduction. I would suggest starting a new paragraph that starts with “Blina is a CD3/CD19-targeting bispecific…” (line 80), and then adding this new text immediately after your summary of the data from TOWER. This will keep all the historical blinatumomab data together.

2.       As formatted, Table 1 is rather confusing. For example, are these the characteristics that were present when the FIRST immunotherapy agent was used? This is what I would assume. However, based on the response to my Major Comment #2, how is it that the median blast percentage prior to starting blinatumomab was 40% if it was only available to those with <5% blasts? Further, are the toxicities reported accumulated through exposures to both immunotherapy agents, or only the FIRST? It was not until reading the end of the Results (page 6, lines 239-245) that I could confirm this was the latter. The right-hand column of Table 1 should presumably be titled “InO/Blina population,” not just “InO.” Please provide more of an explanation of how these data were collected, placing this either in the legend of the Table or in the manuscript text. Following this description, please also address some of the potential errors or discrepancies described above.

3.       Similar to the point raised above in Comment #2, it is stated in the Results (page 5, lines 174-175) that the “median BM blast count was 40%.” Please explain how this is possible if blina was “available in the case of blast <5%” (Patients and Methods, page 3, line 115).

4.       In the paragraph added to describe “intercurrent” therapy (Results, page 5, lines 194-195), it would be more useful to see the median number and range of therapies given instead of the relatively vague description that “more than one treatment [was] possible.” I am also not sure about the use of the word “intercurrent” in this context, which I generally think of meaning “at the same time” or “coincident.” Therefore, there may be confusion about when these therapies were given relative to blina or InO. I would suggest changing this to something like, “Forty pts received chemotherapy/TKI therapy between their exposures to the two immunotherapy agents.”

5. I assume these changes were based on comments from other reviewer(s). However, the first portion of the Discussion (page 6-7, lines 251-287) are now a rather disjointed recitation of previous and ongoing studies using these different agents. There is relatively little attempt to synthesize this information to provide the reader with a clear context into which these new data can be placed. I found this difficult to read.

Some of the additions to the manuscript to address comments from reviewers seem less well-edited than the original submission.

Author Response

Thank you for your comments to which we have tried to answer on a point by point basis.

Query 1: The addition of the data from ALCANTARA is welcome, but it seems out of place where it was added to the Introduction. I would suggest starting a new paragraph that starts with “Blina is a CD3/CD19-targeting bispecific…” (line 80), and then adding this new text immediately after your summary of the data from TOWER. This will keep all the historical blinatumomab data together.

Answer: we have followed Reviewer's suggestion

Query 2: As formatted, Table 1 is rather confusing. For example, are these the characteristics that were present when the FIRST immunotherapy agent was used? This is what I would assume. However, based on the response to my Major Comment #2, how is it that the median blast percentage prior to starting blinatumomab was 40% if it was only available to those with <5% blasts? Further, are the toxicities reported accumulated through exposures to both immunotherapy agents, or only the FIRST? It was not until reading the end of the Results (page 6, lines 239-245) that I could confirm this was the latter. The right-hand column of Table 1 should presumably be titled “InO/Blina population,” not just “InO.” Please provide more of an explanation of how these data were collected, placing this either in the legend of the Table or in the manuscript text. Following this description, please also address some of the potential errors or discrepancies described above.

Answer: in order to avoid any misunderstanding, we have specified in Table 1 the characteristics at first and second immunotheraphy (where available), and we have specified this in the Table itself and in the legend.

Query 3: Similar to the point raised above in Comment #2, it is stated in the Results (page 5, lines 174-175) that the “median BM blast count was 40%.” Please explain how this is possible if blina was “available in the case of blast <5%” (Patients and Methods, page 3, line 115).

Answer: we apologize for the misunderstanding. We have clarifyed the point in the text.

In summary, drug approval for blina in Italy is for hematologic relapse and MRD persistence, while for InO is only for hematologic relapse. Nevertheless, for some patients, InO has been used, in an off labed prescription, also in some cases with bone marrow blast <5%, and we have better specified these aspects in the text.

Query 4: In the paragraph added to describe “intercurrent” therapy (Results, page 5, lines 194-195), it would be more useful to see the median number and range of therapies given instead of the relatively vague description that “more than one treatment [was] possible.” I am also not sure about the use of the word “intercurrent” in this context, which I generally think of meaning “at the same time” or “coincident.” Therefore, there may be confusion about when these therapies were given relative to blina or InO. I would suggest changing this to something like, “Forty pts received chemotherapy/TKI therapy between their exposures to the two immunotherapy agents.”

Answer: we have followed Reviewer's suggestions

Query 5: I assume these changes were based on comments from other reviewer(s). However, the first portion of the Discussion (page 6-7, lines 251-287) are now a rather disjointed recitation of previous and ongoing studies using these different agents. There is relatively little attempt to synthesize this information to provide the reader with a clear context into which these new data can be placed. I found this difficult to read.

Answer: we have tried to semplify this part of the discussion.

Round 3

Reviewer 2 Report

Thank you for addressing my previous comments and questions. 

In the column for "InO/Blina" group in Table 1, bone marrow blast % pre-blina is listed twice. I assume the first one should be changed to pre-InO.

Figure 2 is interesting and helpful. However, I had a hard time understanding what the colors represent in the depiction of response to InO (e.g., brown, pink, purple, etc.). What the other colors were meant to represent was easier to recognize. Consider adding to the Figure Legend a brief description of what these colors are supposed to depict.

As revisions to the Discussion have occurred, it has become more of a list of individual sentences than organized paragraphs. It may be easier to read if this section were reorganized.

Minor editing needed.

Author Response

Thank you for your comments to which we have tried to answer on a point by point basis.

Query 1:

In the column for "InO/Blina" group in Table 1, bone marrow blast % pre-blina is listed twice. I assume the first one should be changed to pre-InO.

Answer: 

we have followed Reviewer's observation

Query 2:

Figure 2 is interesting and helpful. However, I had a hard time understanding what the colors represent in the depiction of response to InO (e.g., brown, pink, purple, etc.). What the other colors were meant to represent was easier to recognize. Consider adding to the Figure Legend a brief description of what these colors are supposed to depict.

Answer: 

we have followed Reviewer's suggestions

Query 3:

As revisions to the Discussion have occurred, it has become more of a list of individual sentences than organized paragraphs. It may be easier to read if this section were reorganized.

Answer: 

we have followed Reviewer's suggestions and tried to make the points more descriptive